# Unlocking the Potential of Circulating miRNAs as Biomarkers in Glioblastoma

**DOI:** 10.3390/life14101312

**Published:** 2024-10-16

**Authors:** Sanika Suvarnapathaki, Antolin Serrano-Farias, Jonathan C. Dudley, Chetan Bettegowda, Jordina Rincon-Torroella

**Affiliations:** 1Department of Neurosurgery, Johns Hopkins University School of Medicine, Baltimore, MD 21218, USA; ssuvarn1@jh.edu (S.S.); aserran7@jh.edu (A.S.-F.); jdudley5@jh.edu (J.C.D.); cbetteg1@jhmi.edu (C.B.); 2Department of Pathology, Johns Hopkins University School of Medicine, Baltimore, MD 21218, USA

**Keywords:** glioblastoma, miRNA, early detection, liquid biopsy, circulating biomarkers, precision medicine

## Abstract

Using microRNAs (miRNAs) as potential circulating biomarkers in diagnosing and treating glioblastoma (GBM) has garnered a lot of scientific and clinical impetus in the past decade. As an aggressive primary brain tumor, GBM poses challenges in early detection and effective treatment with significant current diagnostic constraints and limited therapeutic strategies. MiRNA dysregulation is present in GBM. The intricate involvement of miRNAs in altering cell proliferation, invasion, and immune escape makes them prospective candidates for identifying and monitoring GBM diagnosis and response to treatment. These miRNAs could play a dual role, acting as both potential diagnostic markers and targets for therapy. By modulating the activity of various oncogenic and tumor-suppressive proteins, miRNAs create opportunities for precision medicine and targeted therapies in GBM. This review centers on the critical role and function of circulating miRNA biomarkers in GBM diagnosis and treatment. It highlights their significance in providing insights into disease progression, aiding in early diagnosis, and potential use as targets for novel therapeutic interventions. Ultimately, the study of miRNA would contribute to improving patient outcomes in the challenging landscape of GBM management.

## 1. Introduction

Glioblastomas (GBMs) are the most common and devastating primary malignant tumors of the central nervous system (CNS) in adults and represent 14.2% of all CNS tumors and 50.1% of all malignant brain tumors [1]. The annual incidence rate of GBM is 3.26 cases per 100,000 population, with a median overall survival of 15 months and a 5-year survival rate of 6.9% [1].

GBM is diagnosed through magnetic resonance imaging (MRI) and tissue biopsy [2]. The gold standard for GBM diagnosis is the histopathological analysis and molecular profiling of the biopsied or resected tissue [2]. Both methods for diagnosis and monitoring have significant limitations. For instance, MRI has limitations in its ability to differentiate GBM from other pathological processes (e.g., low-grade gliomas, primary CNS lymphoma, brain abscess, etc.), and it can be challenging to distinguish tumor progression from treatment-related lesions (e.g., pseudoprogression). This is largely attributed to multiple variables affecting the resolution of the image, such as magnetic field distribution inhomogeneity, the spectral resolution of clinical scanners, limited imaging representation of tumor metabolism, and changes in signal intensity based on the location of the tumors [3,4]. Similarly, tissue biopsy is associated with relatively high risks of brain surgery, the known limited representation of tumor heterogeneity, and the inability of real-time tumor evaluation [5]. Moreover, the financial, emotional, and physical toll of repeating highly invasive surgical procedures on patients and the healthcare system also poses a challenge to patients’ overall health and well-being.

### Importance of Liquid Biopsies for GBM Detection

Maximal safe resection is considered a cornerstone of GBM management [6,7]. This improves patient prognosis and facilitates histopathological and molecular analysis [7,8]. Nevertheless, surgical resection is often precluded by the patient’s general condition, tumor location, or multifocality of the lesion [9]. In those cases, a needle biopsy may be an alternative [10]. Although brain tissue biopsy, either through a craniotomy or a needle biopsy, is considered a safe procedure, it carries the potential risk of severe complications that sometimes have to be addressed surgically or may be life-threatening. Moreover, these complications can not only make diagnosis difficult but also complicate patient admissions, lengthen follow-up times, and delay or preclude adjuvant therapy [11]. For example, the potential complication rate for stereotactic biopsy ranges from 7.36% to 28%, with reported mortality ranging from 0% to 3.3% in the literature [10,12,13,14,15,16]. In a retrospective analysis of 1500 brain biopsies, Riche et al. reported that symptomatic complications increased the length of hospitalization and the costs of care in 86.7% of patients to an average extra cost of $35,702 [12].

Given these constraints, liquid biopsy is an emerging diagnostic method capable of detecting brain tumors in cerebrospinal fluid (CSF) or plasma/serum [17]. Circulating biomarkers can be used as a diagnostic tool, decreasing the need for risky brain biopsies and enabling clinicians to diagnose and monitor tumor progression [17].

These span a spectrum of molecules such as circulating tumor DNA (ctDNA), miRNAs, circulating tumor cells (CTCs), and extracellular vesicles (EVs), offering quantitative and qualitative insights into tumor dynamics [18]. Liquid biopsies have demonstrated their utility in providing prognostic information and treatment guidance for various solid malignancies other than GBM [19,20,21,22]. Recent studies have shown that ctDNA was detectable in >75% of patients with advanced pancreatic, ovarian, gastroesophageal, colorectal, bladder breast, melanoma, hepatocellular, and head and neck cancers [22]. Liquid biopsy has also been studied in nervous system sarcomas, such as malignant peripheral nerve sheath tumors or chordomas [23,24]. In gliomas, liquid biopsy’s specificity for biomarker detection in blood is noteworthy, but its sensitivity remains limited [25]. For example, in a recent study, less than 10% of patients with gliomas harbored detectable ctDNA in plasma [22]. While this approach has shown promise, the inherent nature of the blood–brain barrier (BBB) poses significant challenges in detecting ctDNA in circulating blood. For instance, the BBB comprises tightly locked cells that are highly impermeable to foreign substances, essentially shielding the brain from harmful germs, foreign agents, and toxins that could potentially cause damage to the brain. This impermeable nature of the BBB also creates an isolation between the tumor contents and the bloodstream, thus making detection of circulating biomarkers challenging. Theoretically, GBM’s pro-angiogenic and inflammatory microenvironment disrupts the BBB by decreasing tight junctions, leading to tumoral components being shed into the CSF and the bloodstream [17]. While this approach has shown promise, the inherent nature of the BBB poses significant challenges in the detection of these valuable biomarkers in circulating blood. To overcome these challenges, more sensitive detection methods are being explored coupled with techniques such as Magnetic Resonance Imaging-Guided Focused Ultrasound (MRgFUS), which allows temporary, reversible opening of the BBB, allowing more tumor contents to shed into circulating blood. The use of such reversible BBB opening using MRgFUS may increase circulating biomarkers to clinically detectable concentrations, allowing clinical translation of liquid biopsy approaches [26]. Alternatively, the study of biomarkers in CSF instead of blood and plasma may increase the yield of liquid biopsy for brain tumors [22,27,28], especially with novel sequencing strategies based on aneuploidy and the copy number variation being developed [25,29].

Currently, there are no effective interventions for screening CNS tumors apart from imaging techniques such as MRI, CT, PET/SPECT, or surgical biopsies [30]. These methods have limitations in terms of specificity, cost, and invasiveness. Therefore, a significant unmet need exists for developing safer and less-invasive screening methods for CNS tumors. The detection of GBM circulating biomarkers holds promise for improving clinical decision-making, early diagnosis, precise disease progression monitoring, and timely treatment response evaluation [31]. Figure 1 depicts the rationale of the liquid biopsy approach to facilitate the detection of circulating biomarkers through CSF and blood.

## 2. Circulating miRNAs: Definition and Their Role as Biomarkers in Neurological Diseases

miRNAs are small noncoding RNA molecules of ~22 bp average size that regulate gene expression post-transcriptionally by interacting with messenger RNA targets [32]. They are crucial in controlling cellular processes like proliferation, differentiation, and apoptosis. Over the last decade, circulating miRNA found in patients’ CSF, blood, and serum have emerged as promising biomarkers for diagnosing many neurological diseases. For example, some studies have noticed differences in specific miRNAs in the serum of individuals with mild cognitive impairment and Alzheimer’s disease (AD) as compared to control groups. Similarly, the potential diagnostic utility of circulating miRNAs in CSF or the circulation for multiple sclerosis (MS) has been the subject of reports by numerous research groups [33,34,35,36].

### 2.1. Circulating miRNAs and Their Role in Gliomas

These miRNAs have been implicated in both tumor promotion and suppression in gliomas. For instance, miR-155 may stimulate cell growth and suppress cell senescence in gliomas [37]. Conversely, miR-181 and miR-410 have exhibited tumor suppressive roles in gliomas by reducing tumor growth and invasion [38]. These miRNAs can be found circulating in bodily fluids like blood, urine, and CSF, offering potential as diagnostic biomarkers for high-grade gliomas [38]. The critical differences between ctDNA and miRNA as biomarkers lie in their origin. ctDNA is believed to be released passively by dying tumor cells, while miRNA may be actively released by the cancer cells depending on differential cellular activities or status. Therefore, although ctDNA may be a more stable biomarker for early detection and less influenced by biological processes, miRNA may be more effective in depicting the biological makeup of the cancer, including functional state, aggressiveness, or response to therapy [39,40].

Specific miRNAs play pivotal roles in the pathogenesis of GBM, impacting tumor behavior and treatment response. Several miRNAs act as tumor suppressors in GBM. miR-181a and miR-181b, when reduced in expression, correlate with increased tumor malignancy. These molecules inhibit cell proliferation and induce apoptosis, representing potential therapeutic targets. Likewise, reduced miR-34a levels, attributed to epigenetic factors and mutations, affect various oncogenes, hindering cancer progression and invasiveness. When underexpressed, miR-146b-5p, miR-7, miR-124, miR-137, and miR-101 contribute to unregulated mitosis and tumor growth. On the other hand, certain miRNAs, like miR-21, miR-10b, miR-93, miR-196, miR-221, miR-222, and miR-182, act as oncogenic drivers in GBM [41,42]. They facilitate cell proliferation, migration, invasion, and treatment resistance, directly impacting cancer behavior. For instance, miR-21 suppresses genes crucial for apoptosis, such as TIMP3 and RECK genes, while miR-93 influences autophagic activity and regulates neo-angiogenesis [43,44,45].

Understanding the roles of these miRNAs in GBM is crucial for diagnostic and therapeutic advancements.

### 2.2. Role of miRNA in Tumor Suppression

The suppressor miRNAs, particularly miR-181a, miR-181b, miR-34a, miR-146b-5p, miR-7, miR-124, miR-137, miR-101, and miR-128, have been associated with significant roles in glioma development [46,47].

The reduced activity of miR-181a and miR-181b correlates with increased malignancy in glioma cells because they promote increased cell proliferation while suppressing apoptosis. Conversely, overexpression of these miRNAs inhibits proliferation, induces apoptosis, and limits cell invasion, making them potential suppressors in glioma [47,48].

Research has also shown that reduced levels of miR-34a regulate oncogenes like *C-MET* and *NOTCH* [49]. While the precise mechanism causing the dysregulation of miR-34a in human cancer remains incompletely understood, substantial evidence indicates the involvement of an epigenetic process. Transcriptional silencing through CpG methylation emerges as a significant mechanism leading to the deactivation of tumor suppressor genes. Comparable to genomic loss, inactivation due to CpG methylation can facilitate clonal growth, providing an advantage in disease progression. Hypermethylation occurring in the CpG islands within the miR-34a gene promoter region has been observed across a spectrum of solid neoplasms [50]. This hypermethylation blocks cell cycle progression, survival, and invasiveness. miR-34a inhibits Notch signaling, hindering angiogenesis and proliferation [49,51,52].

Similarly, lower expression of miR-146b-5p in gliomas inhibits *EGFR* and the *Pi3K/AKT* pathway, which is potentially valuable for treating invasive cancers. Its regulation impacts *MMP16*, affecting tumor invasion, migration, and blood vessel formation [9,53,54].

With respect to miR-7, miR-124, miR-137, miR-101, and miR-128, each of these miRNAs is associated with inhibiting proliferation, inducing apoptosis, and regulating genes associated with glioma cell growth and invasion [55,56]. For instance, miR-124 and miR-137 lower *CDK6* expression, thereby limiting GBM cell proliferation, while miR-101 downregulates *EZH2*, impacting cell migration and vascularization [57].

These miRNAs, functioning as suppressors, may impact crucial glioma development and progression pathways.

### 2.3. Oncogenic miRNAs in Glioma: A Synopsis of Key Regulators

Several studies report the specific role of some miRNAs that are overexpressed or are found to contribute significantly to the control of the tumor’s metabolism and oncogenic properties. A study by Labib et al. from 2022 revealed that miR-21, highly expressed in various cancers, including glioma, targets crucial genes like *PDCD4, MTAP*, and *SOX5*. qPCR analysis was performed on ctDNA isolated from blood samples collected from patients diagnosed with GBM and healthy patients [58]. The results revealed that the average levels of miR-21 were considerably elevated in the GBM group compared to healthy controls. Conversely, the average expression of miR-181 was reversed. A substantial increase in expression was observed in the control group as opposed to participants with GBM. Notably, miR-21 overexpression is inversely correlated with *PDCD4* expression, hindering apoptosis in glioma cells [9,58].

Another study by Sun L. et al. from 2011 reported that miR-10b is highly oncogenic in GBM and influences targets like *RhoC*, *uPAR*, and *HOXD10* [59]. Inhibition of miR-10b curbs cell growth, invasion, angiogenesis, and boosts apoptosis by regulating *BCL2L11*, *TFAP2C*, *CDKN1A*, and *CDKN2A*. A series of tests were conducted on GBM cell lines U87, LN229, and U251, such as transwell cell invasion analysis, TaqMan^®^ MicroRNA assay-based real-time RT–PCR (Thermofischer Scientific, Carlsbad, CA, USA), and cell apoptosis rate and cell cycle, which were analyzed on FACScan by flow cytometry. The results collectively confirmed that miR-10b played an essential function in GBM cell metabolism, growth, apoptosis, and invasion, indicating that miR-10b could be a potential miRNA biomarker for GBM [9,59].

In a comprehensive study by Huang T. et al. from 2019, elevated miR-93 levels in GBM were found to control cellular functions through targeting *P21* [60]. A large panel of GBM cell lines were analyzed in vitro and in vivo with qRT-PCR, immunoblotting, immunofluorescence staining, glioma sphere formation assay, cell growth, and luciferase reporter assays. This study highlighted the role of miR-93 in regulating the self-renewal of GBM stem cells (GSCs) and the formation of GBM tumors. This was achieved by targeting many important autophagy regulators, namely *BECN1*, *ATG5*, *ATG4B*, and *SQSTM1* [60]. By reducing autophagic activity via ectopic expression of miR-93 or utilizing neural stem cell (NSC) autophagy inhibitors or Chloroquine (CQ), the effectiveness of the commonly used cytotoxic treatments temozolomide (TMZ) and IR radiation in suppressing tumors was increased. The findings of this study indicated that the modulation of these genes not only impacted autophagy but also influenced cell proliferation and the self-renewal of GSCs. These experiments also revealed that the upregulation of miR-196 in GBM cells correlates with reduced overall survival [9,60]. The role of miR-93 in autophagy regulation highlights the potential for a combined treatment approach, wherein autophagy inhibition is coupled with cytotoxic therapy administration [9,60].

Similarly, miR-221 and miR-222 are also increased in glioma. These regulate cell cycle, proliferation, and apoptosis by targeting *p27*, *p57,* and the *PUMA* gene. Their increased expression may hinder programmed cell death [9,61].

miR-182 overexpression escalates with the degree of tumor malignancy in glioma cells, showing a substantial increase in GBM compared to normal brain tissues. This miRNA is coded in the chromosome 7q32.1 region within the *FRA7H* site, and the *MET* gene is often amplified in GBM cells [9,62].

Figure 2 summarizes the reported role played by miRNAs in GBM tumorigenesis and treatment resistance.

### 2.4. Circulating miRNA as Biomarkers for Diagnosis and Treatment of GBM

Circulating miRNAs are relatively stable and can be easily extracted, detected, and quantified, making them valuable biomarkers. They can be susceptible to shedding in the blood, plasma, or CSF when released from tumors via apoptosis, secretions, exocytosis, or extracellular vesicles [63]. Detection of miRNA in blood, plasma, and CSF has gained significant scientific interest due to its potential diagnostic value. While blood/plasma and CSF contain circulating miRNAs, their concentration largely varies in the blood compared to CSF [64]. In addition, systemic factors such as inflammation and other malignancies can skew the miRNA profiles detectable in circulating blood.

On the contrary, CSF may harbor higher concentrations of miRNA due to lower dilution and localized secretion of miRNAs directly in CSF compared to circulating blood [63]. CSF sampling necessitates slightly more invasive techniques for collection via lumbar puncture. However, this is a safe standard procedure employed in the work-up of many neurological diseases. The utilization of miRNA panels from both CSF and blood presents a promising alternative to tissue biopsies, obviating the need for surgical resection of tumor samples solely for diagnostic purposes. Table 1 displays various studies that have discovered possible biomarker miRNAs in the bloodstream and CSF of individuals with GBM.

Every miRNA has one or several known functions in epigenetics, cancer, or regulating progression. miRNAs facilitate cellular adaptation to environmental conditions, enhancing survival during hypoxia and cancer therapies like chemotherapy and radiation [65]. For example, miR-21, typically elevated in GBM patients, may promote tumor development, microvascular proliferation, and resistance to cell death [66,67,68].

**Table 1 life-14-01312-t001:** Consolidated table showing circulating miRNAs in blood, plasma, and CSF, their respective expression levels, specific roles in GBM pathogenesis, and their involvement in various cellular processes and pathways associated with GBM progression [9,69,70].

miRNA	Sample Type	Expression in GBM	Role in GBM	Process/Pathway
miR-182	Blood	Upregulated	oncomiR	Pleiotropic effects on tumor cell growth, invasion, self-renewal, and angiogenesis [71]
miR-21	Blood	Upregulated	oncomiR	Cell cycle, proliferation, and apoptosis [66,67]
miR-146a	Blood	Downregulated	Tumor suppressor	Regulating cell proliferation and differentiation by targeting Notch1 [41]
miR-93	CSF, serum, and tissue biopsy Tissue biopsy	Upregulated	oncomiR	Neo-angiogenesis, anti-inflammatory, controls cellular migration, proliferation, invasion, cell cycle arrest, and chemoresistance by targeting P21 [60]
miR-10b	CSF	Upregulated	oncomiR	Growth and differentiation of CSCs [59]
miR-196a	Blood and plasma	Upregulated	oncomiR	Proliferation and apoptosis [9]
miR-221/222	Blood and plasma	upregulated	oncomiR	Cell cycle, proliferation, and apoptosis [72]
miR-7	CSF, plasma, and serum	Downregulated	Tumor suppressor	Growth and differentiation of CSCs [55]
miR-128	CSF, plasma, and serum	Downregulated	Tumor suppressor	Proliferation and apoptosis [73]
miR-124/137	CSF, plasma, and serum	Downregulated	Tumor suppressor	Growth and differentiation of CSCs [74]
miR-101	CSF, plasma, and serum	Downregulated	Tumor suppressor	Downregulates invasion/proliferation, apoptosis, and migration, by targeting the transcription factor Kruppel-like factor 6 (KLF6) [75]
miR-181	Blood	Downregulated	Tumor suppressor	Proliferation and apoptosis [47]
miR15b	Blood and CSF	Upregulated	Tumor suppressor	Inducing cell cycle arrest [9]
miR-137	Serum	Downregulated	Tumor suppressor	Growth and differentiation of CSCs [76]
miR-34a	Blood and plasma	Downregulated	Tumor suppressor	Cell cycle, Proliferation, and apoptosis [50]

Several additional miRNAs, including miR-10b, miR-106a-5p, miR-185, and miR-210, are elevated in the sera of GBM patients and may be involved in tumor progression [25,65]. Within GBM, the expression levels of miRNAs miR-29, miR-127, miR-137, miR-197, miR-205 are reduced, and these miRNAs act as tumor suppressors [25,65]. miR-221, miR-222, miR-223, and miR-125b-2 influence TMZ responsiveness, whereas miR-128 and miR-301a are involved in GBM’s response to radiotherapy [65,72,77]. Compared to DNA, miRNAs exhibit exceptional stability in blood due to their resistance to RNase degradation and can endure a wide range of storage conditions, including intense pH levels and repeated freezing and thawing [78,79]. miRNAs are found either as free molecules in serum or CSF or enclosed within lipid membranes called exosomes. Exosomes are extracellular membrane vesicles (EVs) with a diameter between 30 and 150 nanometers. They transport specific molecules, including DNA, RNA including miRNA, and proteins, to the recipient cell. Exosomal miRNA loading occurs via lipid-mediated pathways due to the hydrophobic interactions between the miRNA and the lipid membrane of the exosomes [80]. Exosomal miRNAs have become potential biomarkers due to their stability under diverse conditions, such as storage at varying temperatures and excellent cellular permeability [81]. Plasma is rich in EVs, at a density of around 1010 EVs per milliliter [82]. Exosomes have been found to potentially influence tumor development [25,83,84]. They carry a rich combination of RNA, miRNA, proteins, and lipids that reflect their parent cell, remain stable in circulation, and preserve their contents from enzyme breakdown. Moreover, they may traverse the BBB better than other elements [25,85]. Osti et al. showed that glioma patients had higher total circulating exosome concentrations than healthy controls [25,86]. Skog et al. demonstrated that glioma-derived exosomes carried EGFRvIII mRNA in 7 of 25 GBM patients but were absent in healthy individuals [25,87]. Patients had no detectable EGFRvIII two weeks after resection, indicating a direct correlation with tumor burden. Exosomal-derived mRNA has also been proposed to provide a more detailed definition of the genomic features of aggressive cancers such as GBM [25] by mediating O^6^-methylguanine-DNA methyltransferase (MGMT) and alkylpurine-DNA-N-glycosylase (APNG) which repair DNA damage leading to TMZ resistance [88]. Other exosome components, such as noncoding RNA and proteins, could also assist in diagnosis or prognosis [25].

Several reports have revealed the diagnostic importance of different miRNAs. Zhang and colleagues discovered that serum miR-145-5p levels were markedly reduced in participants with GBM compared to controls [89]. miR-145-5p has potential as a prospective diagnostic marker for GBM, with an AUC of 0.89, sensitivity of 84.6%, and specificity of 78.0% in their report [89]. Roth et al. conducted a study that quantitatively examined 1158 mature miRNAs in 20 blood samples from GBM patients [90]. They observed that miR-128 and miR-342-3p were significantly dysregulated in patients with GBM compared to controls. Researchers utilized artificial intelligence algorithms to develop a miRNA profile that achieved 79% sensitivity and 81% specificity in distinguishing blood samples from patients with GBM and healthy participants [90].

When analyzing miRNAs in CSF, miR-15b, miR-21, and miR-1246 can potentially become CSF biomarkers of gliomas. For instance, Baraniskin and colleagues found elevated levels of miR-15b and miR-21 in the CSF of glioma patients when compared to healthy controls [91]. Patients with glioma could be distinguished from both healthy participants and those with primary central nervous system lymphoma with a 90% sensitivity and 100% specificity [91]. Furthermore, a new study found that miR-1246 levels in the CSF of GBM patients are higher than those of low-grade glioma patients [92]. Notably, the concentration of miR-1246 in the CSF of GBM patients decreased after resection [92].

Several exosomal miRNAs may be prognostic biomarkers. A 2013 study found that EVs from CSF in GBM patients had a much higher miR-21 level than healthy controls, with an 87%, sensitivity, 93% specificity, and an AUC of 0.89 [93]. Santangelo et al. found an AUC of 0.80 for a serum panel of exosomal/encapsulated miR-21, miR-222, and miR-124-3p for GBM diagnosis [94]. In addition, patients with high-grade gliomas had significantly lower miRNA expression after tumor excision. A separate study found that exosomes from 75 GBM patients had higher miR-320 and miR-574-3p related to GBM diagnosis when compared to healthy patients in the control group [95].

Histopathological classifications of gliomas may be linked to miRNA signatures. Only three of nine cell-free miRNAs in plasma from 50 GBM patients and healthy controls were significantly dysregulated, as reported in a study by Wang et al. In this study, Wang and colleagues reported that abnormal plasma concentrations of miR-21, miR-128, and miR-342-3p successfully identified biopsy-confirmed GBM individuals compared to healthy controls [96]. For instance, miR-21 was substantially increased in GBM relative to the control group, with an AUC value of 0.93, 90% sensitivity, and 100% specificity. miR-128 and miR-342-3p were substantially reduced with an AUC of 1.00, 90% sensitivity, and 100% specificity in the blood of GBM patients. These researchers also demonstrated an association between miR-342-3p expression levels and the histopathologic classifications of gliomas. Their PCR data showed a noteworthy distinction in the concentration of miR-342-3p among grades 2, 3, and 4 (*p* < 0.008) as the plasma concentration of miR-342-3p demonstrated a discernible decrease in patients with gliomas as tumor grades ascend [96]. In another study, Yue et al. found that glioma patients’ serum miR-205 expression decreased with pathological grade. Expression levels rose after tumor excision and fell after recurrence [97]. Lan et al. found that exosomal miR-301a was elevated in patients with high-grade glioma, while tumor excision decreased it [98]. A 2019 meta-analysis of 47 papers from 20 journals, totaling 2262 glioma patients and 1986 controls, found that cell-free miRNAs accurately identified patients with histologically confirmed glioma in contrast to non-oncologic cohorts with an AUC of 0.91, 83% sensitivity, and 87% specificity. Their sample collection occurred either before or during the surgery. The cell-free miR-21 had the highest pooled AUC of 0.88. MiR-125 and miR-222 were thereafter less effective [99]. They showed again that if the tumor returned, miR-21 expression rose. These findings suggest these miRNAs may be used as non-invasive liquid biopsy biomarkers.

Treatment response surveillance could involve miRNAs. Seigal et al. discovered higher miR-10b and miR-21 levels in bevacizumab-treated GBM patients’ serum [100]. These researchers also found that bevacizumab-treated patients’ tumor sizes were inversely linked with miRNA expression. Monitoring circulating miRNAs may predict therapeutic success because some miRNAs may be dysregulated and reflect the anticancer effect of therapy [100].

GBM prognosis is connected to miRNA dysregulation. Xiao et al. reported that plasma miR-182 expression decreases survival [71]. Zhang and colleagues observed that miR-145-5p correlated with Karnofsky Performance Scale (KPS) scores, IDH1 mutation status, radiation status, and tumor resection [89]. Wang et al. linked low serum miR-485-3p levels to poor prognosis [101]. Lan et al. found a connection between exosomal miR-301a and KPS [98]. Zhang et al. found dysregulated miRNAs in a patient’s serum that may be predictive of survival: increases in miR-20a-5p, miR-106a-5p, and miR-222-3p and decreases in miR-182 and miR-145-5p were substantially associated with shorter 2-year overall survival [102]. Likewise, they found a link between miR-20a-5p and miR-17-5p levels and mortality [102]. Srinivasan et al. constructed a 10-miRNA profile pattern to anticipate GBM survival and found that individuals with shorter or median survival expressed more miR-17-5p, miR-20a, and miR-106 [103]. Contradictory reports may occur from survival term definitions [104]. Yue et al. found that low miR-205 levels correlated with lower KPS [97]. Higher serum miR-205 levels were associated with longer overall survival [87,97]. These findings suggest that specific elevated or decreased miRNA levels may predict GBM prognosis.

Pediatric high-grade gliomas (pHGGs) emit exosomes containing miRNA-1246 and miRNA-1290, which may sustain stemness, cell proliferation, invasion, and chemoresistance in various cancers [105]. Lopez-Aguilar et al. examined serum miRNA levels as non-invasive indicators for pediatric low- and high-grade astrocytoma [106]. They found overexpression of miR-130a and miR-145 relative to controls, which were associated with higher chemotherapeutic response and cell migration and invasion. In blood serum samples, miR-335, a tumor promoter in adult human glioma, was downregulated, suggesting molecular distinctions between adult and pediatric astrocytomas [97].

## 3. Challenges and Research Directions in miRNA-Based Diagnosis and Therapy for GBM

While preliminary results show promise, numerous crucial measures need to be implemented before miRNA can be included in clinical practice. A significant obstacle is the lack of defined methodologies for isolating miRNA, analyzing it, and developing strategies for loading and modification [18,55]. Comprehending the specific function and process of miRNA in cell-to-cell signaling is crucial for investigating their utility for diagnosis and treatment.

The majority of data available to support the role of miRNA as a biomarker are constrained by limited cohort sizes [65]. Thus, further research is necessary to ascertain the sensitivity and specificity of specific miRNAs as biomarkers. Because of the molecular complexity of high-grade gliomas, it will be challenging to detect GBM with one miRNA. A complex molecular signature, including genetic, epigenetic, and miRNA changes, will likely be the key to making liquid biopsy for GBM a reality. In addition, miRNA also exhibits overlap with distinct pathological disorders, and its use will be hindered by sampling differences among different specimens [65]. For instance, miR-21 expression levels have been demonstrated to be elevated in the brain, serum samples, and EVs of individuals diagnosed with GBM but also in various other malignancies, including breast, hepatocellular, colorectal, and ovarian cancers [65]. This discovery implies that, as an example, miR-21 is not exclusive to a particular form of cancer, indicating that a panel of miRNAs may enhance the accuracy of GBM detection and prognosis [65]. This idea has been experimentally validated in studies, wherein simultaneous assessment of miR-15b and miR-21 in CSF demonstrated a robust ability to differentiate between glioma and healthy cohorts and those with CNS lymphomas, achieving 90% sensitivity and 100% specificity [107]. In another study, miR-141, touted as a GBM and prostate cancer differentiator, has demonstrated abnormal expression in other malignancies and even in noncancerous processes [108]. The widely studied miR-21, viewed as a potential indicator of GBM, shows elevated expression in various other cancers, raising concerns about its specificity [108]. Concurrently, assessing the expression levels of miR-15b and miR-21 in CSF enables distinguishing between GBM patients, healthy volunteers, and CNS lymphoma patients with a detection rate of 90% sensitivity and a true negative rate of 100% specificity [108].

Circulating miRNAs possess characteristics that make them potential biomarkers for diverse diseases. Their stability in bodily fluids, whether as ribonucleoprotein complexes or within vesicles, enhances their utility. The ease of detection, facilitated by widely available nucleic acid detection techniques, adds to their clinical potential. The development of new techniques for circulating nucleic acid detection is generally more time and cost-effective than discovering new antibodies for protein biomarkers for several reasons. First, nucleic acids (DNA, RNA) can be detected using relatively straightforward and well-established techniques like PCR and next-generation sequencing, while proteins require complex antibody creation and validation. Second, nucleic acid detection methods are standardized and widely available, whereas developing specific antibodies involves extensive testing to ensure proper specificity and sensitivity. Third, designing primers or probes for nucleic acids can be achieved quickly once the target sequence is known, but antibody discovery involves lengthy processes like immunizing animals and hybridoma creation. Fourth, the costs associated with PCR and sequencing have decreased significantly, making these methods slowly more affordable, whereas antibody development involves higher expenses related to animal care and hybridoma technology. Lastly, high-throughput sequencing technologies allow for the simultaneous analysis of thousands of nucleic acid molecules, whereas high-throughput methods for proteins are generally less efficient and more expensive. However, diagnostic specificity and reproducibility challenges persist, which warrant a more thorough investigation [108].

Ensuring reproducibility is crucial for validating tumor biomarkers. Inconsistent findings, particularly evident in studies on miR-200c and miR-145, highlight challenges in establishing reliable expression patterns. For miR-200c, one study associates its enhanced expression with poor progression and overall survival in gastric cancer patients, while others link it to progression-free survival [109,110]. Similarly, miR-145 is reported to be highly overexpressed in early-stage cancer patients’ plasma compared to healthy individuals across various ethnic groups [111]. However, other studies indicate that miR-145 is downregulated in tissue and plasma samples of breast cancer patients compared to controls, highlighting the variability in its expression [74]. Factors such as limited sample size, poor statistical validity, detection method specificity, and miRNA degradation contribute to result discrepancies. Methodological variations in miRNA extraction techniques and the minimal concentration of circulating miRNAs further complicate reliable quantitation and detection. Therefore, the need for verification in large populations before integrating circulating miRNA profiles into tumor progression and clinical outcomes is emphasized.

Integrating specific miRNAs with traditional tumor markers offers a promising strategy to achieve increased sensitivity and specificity. Additionally, machine learning algorithms, such as support vector machines, highlight the potential of a combination of seven miRNAs (miR-10b, -21, -141, -200a, -200b, -200c, and -125b) in CSF to accurately distinguish GBM and metastatic brain cancer, achieving high accuracy rates ranging from 91% to 99%. Screening such panels of miRNA simultaneously has been shown to offer better insight into miRNA signatures for particular grades of gliomas, which is a prominent line of investigation for improved preclinical and clinical miRNA-based diagnostics [91,112]. Machine learning algorithms could be highly beneficial in detecting prospective miRNA signatures within different bodily fluids from patients with GBM.

Several positive traits have made researchers consider miRNA as a therapeutic for GBM. Their natural capacity to traverse the BBB, biocompatibility, low immunogenic response, and advanced surface-engineering capabilities have made them a prime subject for investigation as vehicles for chemotherapy and nucleotide medications targeting the suppression of gliomas. However, when discussing the use of miRNA as a potential therapeutic against GBM, challenges exist in effectively delivering miRNAs to this type of cancer. Various delivery systems, including nanoparticles, have shown potential success, yet concerns about toxicity and side effects of these approaches, such as immune system activation and off-target effects persist [113]. Polymer and lipid nanoparticles have been extensively used for delivery, with demonstrated success in enhancing sensitivity to chemotherapy [113]. Specifically, passing the highly selective BBB remains a significant challenge that must be addressed to deliver GBM therapeutics effectively [113]. Table 2 summarizes the most pertinent challenges in miRNA-based diagnostics and therapies for treatment of GBM.

It is encouraging that miRNAs have already been used in clinical trials for other pathologies. A clinical trial is investigating *MRX34*, a liposome-formulated miR-34 mimic, for treating patients with advanced solid tumors including liver cancer. The trial aims to assess the safety and efficacy of this miRNA-based therapeutic [114]. In another clinical trial, TargomiRs, which are miR-16-based microRNA mimics, were used to treat patients with malignant pleural mesothelioma. This trial aimed to assess the safety, dosage, and preliminary efficacy of this miRNA therapy [114,115]. The role of miRNA in evaluating cancer pathologies was also studied in a clinical trial evaluating the safety and therapeutic potential of miR-29b for patients with hematologic malignancies like acute myeloid leukemia (AML) and myelodysplastic syndrome (MDS). miR-29b is known to target multiple oncogenes involved in these cancers [116]. Another example of such a trial involves the use of Miravirsen, an antisense oligonucleotide targeting miR-122, for the treatment of chronic hepatitis C virus (HCV) infection, which can lead to liver cancer. The trial aimed to evaluate the safety and efficacy of Miravirsen in reducing HCV RNA levels [117].

Overall, further research is needed to address the challenges of miRNA delivery in GBM, enhance the specificity and sensitivity, and unravel the complex interplay between miRNAs, cancer cells, and treatment resistance [69,70].

**Table 2 life-14-01312-t002:** Challenges and research directions in miRNA-based diagnosis and therapy for GBM.

Challenges	Description	Research Directions	Approach
Heterogeneity of GBM	Genetic, epigenetic, and phenotypic variability complicates the identification of universal miRNA biomarkers and therapeutic targets	Biomarker Discovery	Advancing high-throughput sequencing and bioinformatics approaches for reliable miRNA biomarker discovery [118]
Delivery Mechanisms	Efficiently delivering miRNA-based therapeutics across the blood–brain barrier (BBB) remains a challenge	Improved Delivery Systems	Developing novel delivery platforms (e.g., exosomes, targeted nanoparticles, advanced viral vectors) to enhance delivery [119]
Off-target Effects	miRNAs can target multiple mRNAs, leading to potential off-target effects and unintended consequences	Therapeutic Specificity	Ensuring specificity in miRNA-based therapies to minimize adverse effects [120]
Stability and Degradation	miRNAs are prone to degradation by nucleases in the bloodstream	Stable Therapeutic Agents	Developing stable miRNA mimics or inhibitors that retain functionality [57]
Tumor Microenvironment	The complex GBM microenvironment can influence the efficacy of miRNA-based therapies	Microenvironment Interaction	Understanding and manipulating interactions within the tumor microenvironment to optimize therapeutic outcomes [121]
Resistance Mechanisms	GBM cells can develop resistance to miRNA-based therapies through various mechanisms	Overcoming Resistance	Identifying and overcoming resistance mechanisms for long-term therapeutic success [121]
Combination Therapy Needs	Combining different therapeutic approaches can enhance efficacy and reduce resistance	Combination Therapies	Exploring synergistic effects of combining miRNA-based therapies with chemotherapy, radiotherapy, and immunotherapy [122]
Understanding miRNA Functions	Comprehensive understanding of miRNA roles in GBM biology is needed	Functional Studies	Conducting studies to elucidate specific miRNA roles in tumor growth, invasion, and microenvironment interactions [9]
Clinical Validation	Ensuring the safety and efficacy of miRNA-based diagnostics and therapeutics in clinical settings	Clinical Trials	Designing and conducting trials to evaluate miRNA-based diagnostics and therapies in GBM patients [9]
Personalized Treatment	Tailoring treatments to the unique genetic and molecular landscape of each patient’s tumor	Personalized Medicine	Leveraging miRNA profiling for individualized treatment strategies [9]
Regulatory Compliance	Establishing standards for safety, efficacy, and reproducibility in miRNA-based applications	Regulatory Frameworks	Developing clear guidelines and standardization protocols for miRNA-based diagnostics and therapies [69]

## 4. Conclusions and Future Directions

Investigating circulating miRNAs as potential diagnostic and therapeutic biomarkers for GBM holds promise. Prior studies reveal their significance in understanding glioma genesis, offering insights into tumor progression, and perhaps in the future, aiding in clinical decision-making [41,91]. Circulating miRNAs, particularly those present in CSF, show potential as valuable early detection biomarkers.

Efforts are underway to develop comprehensive miRNA panels to improve sensitivity and specificity in screening and diagnosis. For example, the identification of specific miRNA expression patterns has exhibited high sensitivity and specificity in differentiating patients with GBM from noncancer individuals, offering a potential diagnostic model with promising accuracy [55]. Technologies such as MRgFUS, which enhance blood–brain barrier permeability, could also aid in the detection efficacy and concentrations of these miRNAs in blood, thus further advancing the diagnostic and therapeutic potential of these miRNA panels [123].

Despite the promising prospects of miRNA-based approaches, several challenges need to be addressed to translate these findings into clinical practice. One major obstacle is the standardization of miRNA detection methods. Variability in miRNA quantification can impact the reproducibility of results, which is essential for their reliable application in diagnostics [120]. Furthermore, the heterogeneity of GBM complicates the identification of universal miRNA biomarkers, as expression profiles can vary significantly between individual tumors and patient populations [124]. To overcome these issues, refined profiling techniques and standardized protocols for miRNA isolation and quantification are needed.

Advancements in high-throughput sequencing and quantitative PCR have enhanced the sensitivity and specificity of miRNA detection, paving the way for more accurate diagnostics [98]. Bioinformatics tools are also playing a crucial role in analyzing miRNA expression patterns and identifying potential biomarkers with high diagnostic and therapeutic value.

Therapeutic strategies involving miRNAs are also evolving. The use of miRNA mimics to restore the function of downregulated tumor-suppressive miRNAs, such as miR-34a, has shown promise in preclinical models by enhancing tumor suppression and improving responses to conventional therapies [49]. Conversely, targeting oncogenic miRNAs like miR-21 with antagomirs or locked nucleic acid (LNA) technology has demonstrated potential to reduce tumor malignancy and improve treatment outcomes [125]. Furthermore, the development of novel delivery systems for miRNA-based therapeutics is addressing the challenge of effective and targeted delivery. Nanoparticle-based delivery systems, for instance, offer the potential to enhance the stability, bioavailability, and targeted delivery of miRNA mimics and inhibitors while minimizing off-target effects [89].

By combining MRgFUS with MRI-guided delivery systems, one could visualize and track the distribution and effectiveness of miRNA mimics or inhibitors in real time, optimizing the dosing and delivery strategies [123].

In summary, the integration of miRNAs into GBM diagnostics and therapeutics could represent an advancement in personalized medicine. Addressing current challenges, such as standardization of detection methods, validation in diverse patient cohorts, and development of effective delivery systems, will be crucial for realizing the full potential of miRNA-based strategies.

## Figures and Tables

**Figure 1 life-14-01312-f001:**
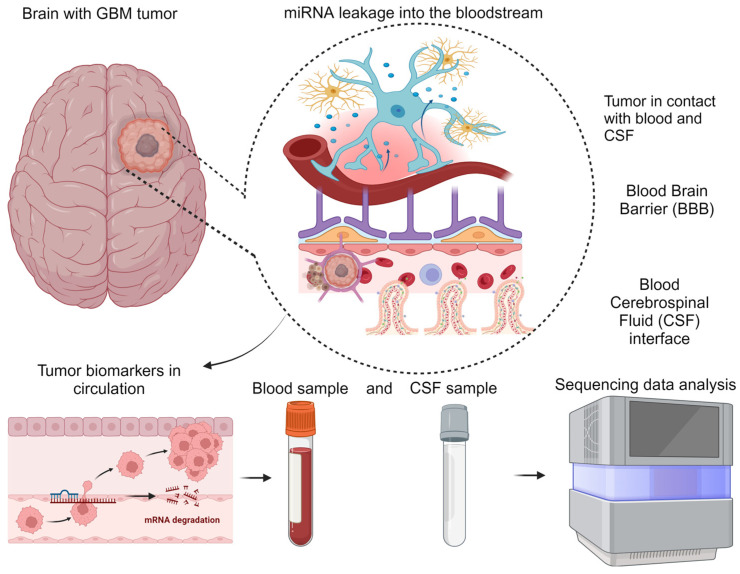
Liquid biopsies present a new diagnostic paradigm enabling less-invasive detection of circulating tumor biomarkers in blood and CSF of patients.

**Figure 2 life-14-01312-f002:**
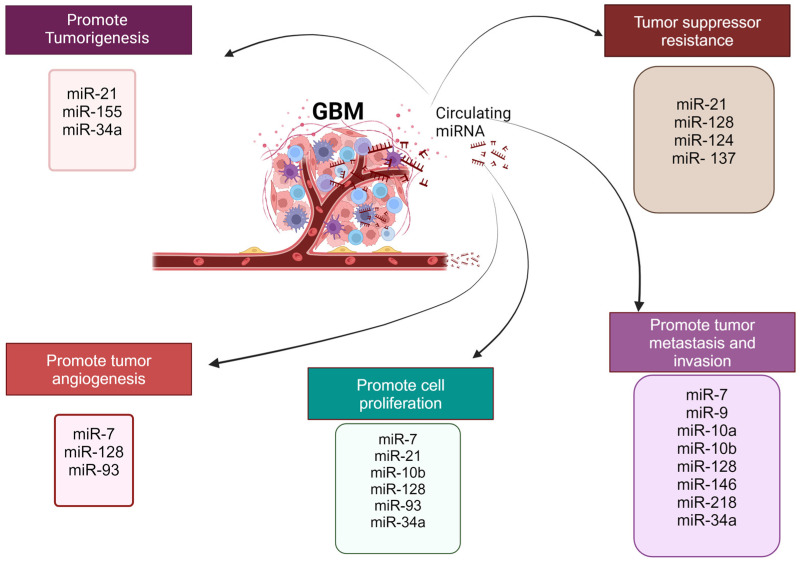
Key roles played by miRNAs in GBM tumor progression, metastasis, and treatment resistance are summarized here, making these miRNAs valuable diagnostic and therapeutic biomarkers for GBM patients.

## Data Availability

No new data were created or analyzed in this study.

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
