# Peer review of "Unlocking the Potential of Circulating miRNAs as Biomarkers in Glioblastoma"

_life, 2024, doi:10.3390/life14101312_

Round 1
Reviewer 1 Report
Comments and Suggestions for Authors
Overall, I thought that this was an excellent summary of the existing information available for miRNA as a potential biomarker for GBM.
I thought that the introduction (Section 1.2) would have benefited from a more elaborate explanation of the role of the blood-brain barrier, the challenges of finding ctDNA because of this and how miRNA is different and a reasonable target for liquid biopsy. If there are references that could be added to support the accuracy of the relationship between circulating miRNA and tumor burden, that would be useful. It would also be interesting to understand the relative sensitivities of detection between finding miRNA in the serum versus the CSF. Finally, I think it would be valuable for the authors to comment on the fact that there are currently no screening tests available for CNS tumors and that there is an unmet need in the emerging field of liquid-biopsy to be able to screen for CNS tumors.
Section 2.4
Please provide more information about the relative concentrations of miRNA in circulation versus in the CSF and how this does or does not effect the accuracy of detecting the presence of disease using this method.
Author Response
The responses to the reviewer’s comments are summarized here and in the attached word document.
We thank the reviewers for their constructive feedback and have addressed every comment point by point. The changes are also incorporated in the revised manuscript highlighted within the manuscript for your convenience. We have made some text edits throughout the manuscript which have also been highlighted in the resubmitted version.
Comment 1: Overall, I thought that this was an excellent summary of the existing information available for miRNA as a potential biomarker for GBM.
I thought that the introduction (Section 1.2) would have benefited from a more elaborate explanation of the role of the blood-brain barrier, the challenges of finding ctDNA because of this and how miRNA is different and a reasonable target for liquid biopsy. If there are references that could be added to support the accuracy of the relationship between circulating miRNA and tumor burden, that would be useful. It would also be interesting to understand the relative sensitivities of detection between finding miRNA in the serum versus the CSF. Finally, I think it would be valuable for the authors to comment on the fact that there are currently no screening tests available for CNS tumors and that there is an unmet need in the emerging field of liquid-biopsy to be able to screen for CNS tumors.
Response: Thank you for your comment and input. We have accepted the reviewer’s suggestions and included the following text in Section 1.2 to discuss these points.
This text is highlighted within the manuscript and is added on page 2 lines 72-74 and 77-96
Recent studies have shown that ctDNA was detectable in >75% of patients with advanced pancreatic, ovarian, gastroesophageal, colorectal, bladder breast, melanoma, hepatocellular, and head and neck cancers [22] .
For example, in a recent study, less than 10% of patients with gliomas harbored detectable ctDNA in plasma [22]. While this approach has shown promise, the inherent nature of the blood-brain barrier (BBB) poses significant challenges in detecting ctDNA in circulating blood. For instance, the BBB is comprised of tightly locked cells that are highly impermeable to foreign substances, essentially shielding the brain from harmful germs, foreign agents, and toxins that could potentially cause damage to the brain. This impermeable nature of the BBB also creates an isolation between the tumor contents and the bloodstream, thus making detection of circulating biomarkers challenging. Theoretically, GBM’s proangiogenic and inflammatory microenvironment disrupts the BBB by decreasing tight junctions, leading to tumoral components being shed into the CSF and the bloodstream [17]. While this approach has shown promise, the inherent nature of the BBB poses significant challenges in the detection of these valuable biomarkers in circulating blood. To overcome these challenges, more sensitive detection methods are being explored coupled with techniques such as Magnetic Resonance Imaging Guided Focused Ultra-sound (MRgFUS), which allows temporary, reversible opening of the BBB, allowing more tumor contents to shed into circulating blood. The use of such reversible opening of the BBB using MRgFUS may enable an increase in concentrations of circulating biomarkers to clinically detectable concentrations, allowing clinical translation of liquid biopsy approaches [26].
This text is highlighted within the manuscript and is added on and Page 3 on Lines 100-104 respectively.
Currently, there are no effective interventions for screening CNS tumors apart from imaging techniques such as MRI, CT, and PET/SPECT, or surgical biopsies [30]. These methods have limitations in terms of specificity, cost, and invasiveness. Therefore, there is a significant unmet need for the development of safer and less invasive screening methods for CNS tumors.
Reference added:
- Korte, B. and D. Mathios, Innovation in Non-Invasive Diagnosis and Disease Monitoring for Meningiomas. Int J Mol Sci, 2024. 25(8).
Comment 2: Section 2.4
Please provide more information about the relative concentrations of miRNA in circulation versus in the CSF and how this does or does not effect the accuracy of detecting the presence of disease using this method.
Response: Thank you for this comment. We have added text in section 2.4 to discuss this point. This text is highlighted within the manuscript in Section 2.4 on Page 6 Lines 244-260 and Page 7 and 8 Lines 311-320 of the manuscript.
Circulating miRNAs are relatively stable and can be easily extracted, detected, and quantified, making them valuable biomarkers. Those can be susceptible to shedding in the blood, plasma, or CSF when released from tumors via processes such as apoptosis, secretions, exocytosis or via extracellular vesicles [63]. Detection of miRNA in blood, plasma, and CSF has gained significant scientific interest due to its potential diagnostic value. While both blood/plasma and CSF contain circulating miRNAs their concentration largely varies in the blood compared to CSF [64]. In addition, systemic factors such as inflammation and other malignancies can largely skew the miRNA profiles detectable in circulating blood. On the contrary, CSF may harbor higher concentrations of miRNA due to lower dilution and localized secretion of miRNA directly in CSF compared to circulating blood [63]. CSF sampling necessitates slightly higher invasive techniques for collection via lumbar puncture. However, this is a safe standard procedure employed in the work-up of many neurological diseases. The utilization of miRNA panels from both CSF and blood presents a promising alternative to tissue biopsies, obviating the need for surgical resection of tumor samples solely for diagnostic purposes.
When analyzing miRNAs in CSF, miR-15b, miR-21, and miR-1246 can potentially become CSF biomarkers of gliomas. For instance, Baraniskin and colleagues found elevated levels of miR-15b and miR-21 in the CSF of glioma patients when compared to healthy controls [84]. Patients with glioma could be distinguished from both healthy participants and those with primary central nervous system lymphoma with a 90% sensitivity and 100% specificity [84]. Furthermore, a new study found that miR-1246 levels in the CSF of GBM patients are higher than those of low-grade glioma patients [85]. Notably, the concentration of miR-1246 in the CSF of GBM patients decreased after resection [85].
Reference added:
- Loo, H.K., et al., Circulating biomarkers for high-grade glioma. 2019, Taylor & Francis. p. 161-165.
- Seršić, L.V., et al., Real-time PCR quantification of 87 miRNAs from cerebrospinal fluid: miRNA dynamics and association with extracellular vesicles after severe traumatic brain injury. International Journal of Molecular Sciences, 2023. 24(5): p. 4751
84. Baraniskin, A., et al., Identification of microRNAs in the cerebrospinal fluid as biomarker for the diagnosis of glioma. Neuro-oncology, 2012. 14(1): p. 29-33.
85. Qian, M., et al., Hypoxic glioma-derived exosomes deliver microRNA-1246 to induce M2 macrophage polarization by targeting TERF2IP via the STAT3 and NF-κB pathways. Oncogene, 2020. 39(2): p. 428-442.

Reviewer 2 Report
Comments and Suggestions for Authors
The authors wrote a review article about miRNA and glioblastoma. I raised several points that need to be edited.
1. The following sentence is overstatement. “For instance, MRI does not efficaciously differentiate GBM from other pathological processes (e.g., low-grade gliomas, primary CNS lymphoma, brain abscess, etc.), and it cannot discriminate tumor progression from treatment-related lesions (e.g., pseudoprogression)[3].” Indeed, MRI has some limitations, “does not” and “cannot” is not appropriate to use in these points. The authors have to tone down.
2. Table 1 is not organized. I recommend changing the height of each column to minimize the space.
3. Again, Table 2 is not organized. Also, it has different line styles from Table 1.
4. It will be better if Tables 1 and 2 have reference numbers.
5. Gene names need to be written in Italic. (ex. Line 455 PTEN, PDCD4)
6. Line 397, “and off-target effects, persist. persist [98].” Is this a misspelling?
7. There needs to be a space before the bracket of citations. Ex. Line 410, “therapy[99].” Please change them throughout the text.
8. This review does not have any figures. Usually, review articles have nice figures that capture the overview of miRNA and glioblastoma.
Author Response
The responses to the reviewer’s comments are summarized here and also in the response to reviewer's letter attached as a word document. Please see attachment.
We thank the reviewers for their constructive feedback and have addressed every comment point by point. The changes are also incorporated in the revised manuscript highlighted within the manuscript for your convenience. We have made some text edits throughout the manuscript which have also been highlighted in the resubmitted version.
Comment 1: The following sentence is overstatement. “For instance, MRI does not efficaciously differentiate GBM from other pathological processes (e.g., low-grade gliomas, primary CNS lymphoma, brain abscess, etc.), and it cannot discriminate tumor progression from treatment-related lesions (e.g., pseudoprogression)[3].” Indeed, MRI has some limitations, “does not” and “cannot” is not appropriate to use in these points. The authors have to tone down.
Response: Thank you for this comment. We have accepted the reviewer’s suggestion and rewritten the sentence to tone it down as suggested by the reviewer.
The rewritten sentence is highlighted in the manuscript and included on Page 1 Lines 35-42.
For instance, MRI has limitations in its ability to differentiate GBM from other pathological processes (e.g., low-grade gliomas, primary CNS lymphoma, brain abscess, etc.), and it can be challenging to distinguish tumor progression from treatment-related lesions (e.g., pseudoprogression). This is largely attributed to multiple variables affecting resolution of the image such as magnetic field distribution inhomogeneity, spectral resolution of clinical scanners, limited imaging representation of tumor metabolism, and changes in signal intensity based on the location of the tumors [3, 4].
Reference added:
- Galldiks, N., et al., Challenges, limitations, and pitfalls of PET and advanced MRI in patients with brain tumors: A report of the PET/RANO group. Neuro-Oncology, 2024. 26(7): p. 1181-1194
Comment 2: Table 1 is not organized. I recommend changing the height of each column to minimize the space.
Response: Thank you for bringing this to our attention. We have adjusted the height of each column of Table 2 as suggested by the reviewer.
Comment 3: Again, Table 2 is not organized. Also, it has different line styles from Table 1.
Response: Thank you. We have reformatted Tables 1 and 2 according to the reviewer’s recommendations.
References added:
- Arbatskiy, M., et al., Intratumoral Cell Heterogeneity in Patient-Derived Glioblastoma Cell Lines Revealed by Single-Cell RNA-Sequencing. International Journal of Molecular Sciences, 2024. 25(15).
- Singh, R.R., et al., Engineered smart materials for RNA based molecular therapy to treat Glioblastoma. Bioactive Materials, 2024. 33: p. 396-423.
- Godoy, P.M., et al., Comparison of Reproducibility, Accuracy, Sensitivity, and Specificity of miRNA Quantification Platforms. Cell Rep, 2019. 29(12): p. 4212-4222 e5.
- White, J., et al., The tumour microenvironment, treatment resistance and recurrence in glioblastoma. Journal of Translational Medicine, 2024. 22(1): p. 1-14.
- Sandhanam, K., et al., Unlocking novel therapeutic avenues in glioblastoma: Harnessing 4-amino cyanine and miRNA synergy for next-gen treatment convergence. Neuroscience, 2024. 553: p. 1-18.
Comment 4: It will be better if Tables 1 and 2 have reference numbers.
Response: Thank you. We have added refences in Tables 1 and 2 ON PAGES 11,12, and 13 of the manuscript as suggested by the reviewer.
Comment 5: Gene names need to be written in Italic. (ex. Line 455 PTEN, PDCD4)
Response: Thank you for this comment. We have ensured all gene names are italicized as mentioned by the reviewer. The gene names are also highlighted throughout the manuscript were formatted in italic.
Comment 6: Line 397, “and off-target effects, persist. persist [98].” Is this a misspelling?
Response: Thank you for this comment. We want to clarify and confirm that this was an accidental repetition error. We have fixed this error by deleting the repeated word persist.
Comment 7: There needs to be a space before the bracket of citations. Ex. Line 410, “therapy[99].” Please change them throughout the text.
Response: Thank you for bringing this to our attention. We have fixed this spacing issue throughout the manuscript.
Comment 8: This review does not have any figures. Usually, review articles have nice figures that capture the overview of miRNA and gliob
Response: Thank you for this comment. We have now included 2 figures that summarize the role of miRNA as a valuable diagnostic biomarkers for treatment of GBM. The figures have been highlighted within the manuscript.
These changes are added on Page 3 Lines 107 to line 116 and on Pages 6 Lines 235 to Line 241
Figure 1. depicts the rationale of the liquid biopsy approach and how the tumor’s proximity the blood brain and the blood CSF barriers facilitates detection of circulating biomarkers through sampling of blood and CSF.
Figure 1. Liquid biopsies present a new diagnostic paradigm enabling less invasive detection of circulating tumor biomarkers in blood and CSF of patients. (see attached word file for figures)
Page 6 Line 235-241
(see attached word file for figures)
Figure 2. represents the role played by miRNA in GBM tumorigenesis and treatment resistance.
